# HIF1A Knockout by Biallelic and Selection-Free CRISPR Gene Editing in Human Primary Endothelial Cells with Ribonucleoprotein Complexes

**DOI:** 10.3390/biom13010023

**Published:** 2022-12-22

**Authors:** Camilla Blunk Brandt, Sofie Vestergaard Fonager, János Haskó, Rikke Bek Helmig, Søren Degn, Lars Bolund, Niels Jessen, Lin Lin, Yonglun Luo

**Affiliations:** 1Department of Biomedicine, Aarhus University, 8000 Aarhus, Denmark; 2Steno Diabetes Center Aarhus, Aarhus University Hospital, 8200 Aarhus, Denmark; 3Department of Molecular Biology and Genetics, Aarhus University, 8000 Aarhus, Denmark; 4Department of Obstetrics and Gynecology, Aarhus University Hospital, 8200 Aarhus, Denmark; 5Lars Bolund Institute of Regenerative Medicine, Qingdao-Europe Advanced Institute for Life Sciences, BGI-Qingdao, BGI-Shenzhen, Qingdao 266555, China

**Keywords:** endothelial cells, human umbilical vein endothelial cells, HIF1A, hypoxia inducible factor 1 alpha, CRISPR-Cas, gene editing, ribonucleoprotein, non-viral gene editing, transfection, nucleofection

## Abstract

Primary endothelial cells (ECs), especially human umbilical vein endothelial cells (HUVECs), are broadly used in vascular biology. Gene editing of primary endothelial cells is known to be challenging, due to the low DNA transfection efficiency and the limited proliferation capacity of ECs. We report the establishment of a highly efficient and selection-free CRISPR gene editing approach for primary endothelial cells (HUVECs) with ribonucleoprotein (RNP) complex. We first optimized an efficient and cost-effective protocol for messenger RNA (mRNA) delivery into primary HUVECs by nucleofection. Nearly 100% transfection efficiency of HUVECs was achieved with EGFP mRNA. Using this optimized DNA-free approach, we tested RNP-mediated CRISPR gene editing of primary HUVECs with three different gRNAs targeting the HIF1A gene. We achieved highly efficient (98%) and biallelic HIF1A knockout in HUVECs without selection. The effects of HIF1A knockout on ECs’ angiogenic characteristics and response to hypoxia were validated by functional assays. Our work provides a simple method for highly efficient gene editing of primary endothelial cells (HUVECs) in studies and manipulations of ECs functions.

## 1. Introduction

Endothelial cells (ECs) form a single layer of cells lining the vascular systems [1]. Blood ECs are vital conduits that play essential roles in, e.g., the delivery of nutrients and oxygen to the tissues, regulation of immune responses, and regulation of vascular tone [2,3,4], whereas lymphatic ECs which delineate lymphatic vessels are important for immune responses and maintenance of vessel integrity [5]. The ECs are highly heterogeneous, depending on the vascular beds, tissue types (even within a single organ), physiology states, and diseases [6,7,8,9]. ECs from mature tissues or organs mostly stay in a quiescent state but remain metabolically active and can form new vessels through the process termed angiogenesis [4,6]. It is thus not surprising that ECs are involved in many prevalent diseases, including conditions characterized by excess ECs growth (e.g., cancers, eye diseases) or ECs dysfunction (e.g., diabetes, cardiovascular disease) [4].

Cultured ECs are widely used in vascular cell biology and for studies of ECs functions/dysfunctions. Genetic modification of cultured cells is a widely used method for studying biological processes, but the approach has been difficult to apply in primary ECs [10]. Small interfering RNA (siRNA) and short hairpin RNA (shRNA) approaches have been successfully used to knock down the expression of specific genes in ECs [11,12]. However, these approaches are limited by their incomplete or transient gene depletion [13]. Permanent gene disruption by the CRISPR-Cas system is a powerful tool for easy and highly efficient modification of eukaryotic genomes. CRISPR gene editing in primary cells is still challenging due to the limited life span of the primary cells, including primary ECs [14,15,16]. Primary ECs are very limited in replicative life span [17], have low transfection efficiencies, and cannot form single cell-derived colonies [18,19,20]. Hence, most gene editing experiments are performed on immortalized ECs, which attempt to mimic primary ECs but compromise ECs gene expression, phenotype, and other functional characteristics [21,22].

Currently, successful CRISPR-Cas9 gene editing has been reported in primary ECs, with efficiencies ranging from 40% to 80% depending on the used ECs model and CRISPR-Cas9 methods [10,15,23,24,25]. Schwefel et al. previously showed that the gene editing efficiency of immortalized HUVECs with ribonucleoprotein (RNP) was equal to that achieved with lentivirus-based delivery [22]. Viral-free and DNA-free delivery, like RNP, directly delivers CRISPR components into cells and is considered to be safer and more efficient [26]. Indeed, the RNP delivery system has emerged as the most powerful method for CRISPR gene editing due to its advantages of transient expression, no integration, high efficiency, low toxicity, and most importantly reduced off-target effects [27,28]. Promisingly, the RNP system is highly efficient in other primary cells like human lung fibroblasts [29], skin fibroblasts [30], and mature primary mouse innate lymphocyte cells [31]. To our knowledge, very few attempts of RNP-mediated CRISPR-Cas9 gene editing have been performed and validated on primary ECs. Establishing such an RNP-based highly efficient CRISPR gene editing for primary ECs is greatly needed. We hypothesize that RNP will be an efficient method for gene editing of primary ECs, due to its low toxicity effect and high gene editing efficiencies. In this study, we report the establishment and optimization of a highly efficient and selection-free CRISPR method for targeted gene editing of primary ECs (HUVECs). For proof-of-concept, we selected the hypoxia-inducible factor 1-alpha (*HIF1A)* gene, which plays an important role in endothelial adaptation, to establish the method.

## 2. Materials and Methods

### 2.1. Isolation and Culturing of Primary HUVECs

All umbilical cords used for isolation of HUVECs were donated from the Department of Obstetrics and Gynecology, Aarhus University Hospital, Denmark. The isolated HUVECs were cultured on 0.1% gelatin-coated culturing flasks with filters (Nunc, #156499) in M199 medium (Gibco, #22350-029) supplemented with 2 mM L-glutamine (Gibco, #35050-061), 0.4% Endothelial Cell Growth Supplement (ECGS)/Heparin (PromoCell, #C-30120), 20% fetal bovine serum (FBS) (Sigma, Kawasaki, Japan, #F7524) in a 5% CO_2_ atmosphere at 37 °C. The M199 medium with supplements was replaced three times per week. HUVECs were split every 14 days, the trypsination (Gibco, #25300054) was stopped by M199 medium with supplements. The cells were centrifuged at 250× *g* for 3 min and seeded at 5–6 × 10^4^ cells per cm^2^ in 150 µL per cm^2^ combined medium of 50% M199 medium with supplements and 50% endothelial cell growth medium (PromoCell, Heidelberg, Germany, #C-22011). All materials were thoroughly sterilized, and all reagents were filtered through 0.20 µm filters (Sarstedt, #00256103) to avoid contamination. The HUVECs used in these experiments are mycoplasma free (Eurofins Genomics, Ebersberg, Germany, ISO17025 accreditation), CD45- and CD31+ (Appendix A), and have an uptake of low density lipoproteins (Appendix A) which are characteristics of functional ECs. HUVECs were always used as single-donor cultures between passage (p.) 1 and 5.

### 2.2. Preparation of EGFP mRNA and EGFP Plasmid

In vitro transcription (IVT) EGFP plasmid was made by associate professor Rasmus O. Bak, Department of Biomedicine, Aarhus University. EGFP mRNA was generated by IVT. First, the EGFP plasmid was linearized by mixing 15 µL nuclease-free water (Thermo Scientific, Waltham, MA, USA, #R0582), 2 µL 10× Fast Digest Green Buffer (Thermo Scientific, #00959802), 2 µL (1 µg/µL) EGFP plasmid, and 2 µL Fast Digest restriction enzyme BbsI (Thermo Scientific, #00986235) and stored on ice. The reaction mixture was incubated at 37 °C for 3 h. Next, the linearization of the EGFP plasmid was visualized on a 1% agarose gel with a 1 kb marker (Thermo Scientific, #SM0311). NucleoSpin Gel and PCR Clean-up (Macherey, #2006/001) was used to elute the EGFP DNA from the agarose gel by following the manufacturer’s protocol. Finally, IVT was performed by using MEGAscript kit (Thermo Fischer Scientific, #AMB13345) following manufacturer’s protocol with one improvement as 3 µL CleanCap AG (6 mM) (TriLink Biotechnologies, San Diego, CA, USA, #N-7113-5) was added directly to the reaction mix to increase the stability of the mRNA and its translation. The RNA concentration was measured by a Nanodrop 1000 Spectrophotometer. The EGFP mRNA was stored at −20 °C.

### 2.3. Nucleofection of HUVECs with EGFP mRNA Titration

First, the HUVECs were split as described above and incubated for 48 h in a 5% CO_2_ atmosphere at 37 °C. After 48 h incubation, the M199 medium with supplements was changed. The cells were incubated until optimal confluency of 90% before nucleofection.

Preparations for the nucleofection: All reagents were filtered through 0.20 µm filters. The nucleocuvette (Lonza, Basel, Switzerland, #1080781) was thoroughly washed. 24-well culture plates (Sarstedt, #0022721) were coated with 0.1% gelatin. The coated plates were filled with 50% M199 medium with supplements, 50% endothelial cell growth medium, and additional 1% penicillin/streptomycin (Gibco, #15140122), and incubated in a 5% CO_2_ atmosphere at 37 °C until use.

Nucleofection: The 90% confluent HUVECs were divided into six groups (0.0 µg EGFP mRNA (control), 0.4 µg EGFP mRNA, 0.8 µg EGFP mRNA, 1.6 µg EGFP mRNA, 3.2 µg EGFP mRNA, and 4.8 µg EGFP mRNA) with 2.4 × 10^5^ cells in each group. The cells were gently washed twice in PBS, centrifuged at 500× *g* for 5 min, and resuspended in 60 µL OptiMem (Gibco, #11058-021). EGFP mRNA was thawed on ice. The given concentration of EGFP mRNA was added to the appropriate groups and mixed thoroughly by pipetting. Each test group was divided into three wells in the nucelocuvette. Immediately upon transfer to the nucleocuvette, the HUVECs were nucleofected by the 4D-Nucleofector X Unit (Lonza, CH) with the nucleofection program: CM138. After nucleofection, 150 µL prewarmed M199 medium with supplements was added to each well in the nucelocuvette. Finally, the nucleofected HUVECs were seeded in the prepared 24-well plate with a total volume of 500 µL culturing medium and incubated for 24 h in a 5% CO_2_ atmosphere at 37 °C.

Visualization: After 24 h of incubation the expression of EGFP was visualized by fluorescent microscopy and quantified by FC analysis. The cells were visualized by a Leica DMi1 fluorescent microscope, DE, obtaining three brightfield and FITC microscopy images per well at 10× magnification, with the same laser intensities and camera exposures. Image visualization and analysis was done in Fiji (ImageJ) v. 2.9.0. The background signal was adjusted based on averaging background values from four areas negative for signal in each channel and subtracting the mean values from the final image. NovoCyte Quanteon 4025 flow cytometer was used to quantify the percentage of EGFP positive cells. The HUVECs were washed in PBS twice, trypsinized, and centrifuged at 400× *g* for 5 min. The cell pellet was washed twice in PBS + 5% FBS and resuspended in 200 µL PBS + 5% FBS. All samples were incubated in the dark on ice until FC analysis: 100 µL of each sample was acquired and EGFP detected off the 488 nm laser (100 mW) in the B530/30 detector. The FC data were analyzed in NovoExpress v. 1.5.6 using the following gating strategy: ① A forward-scatter-area to side-scatter-area density plot was made to exclude debris, ② A forward-scatter-height to forward-scatter-area density plot was followed by a side-scatter-height to side-scatter-area density plot to exclude doublets, ③ A single parameter histogram was made to identify the cells expressing EGFP. This experiment was performed with one HUVECs doner at p. 5 (*n* = 3).

### 2.4. Nucleofection of HUVECs with EGFP mRNA and EGFP Plasmid

The nucleofection of HUVECs protocol, described above was used to nucleofect HUVECs with 3.2 µg EGFP mRNA or 1.312 µg EGFP plasmid. The concentration EGFP plasmid had to be lower than EGFP mRNA since high plasmid concentrations are toxic to the cells. This experiment was repeated to achieve both biological- and technical triplicates, by using three different HUVECs donors at p. 2–5.

### 2.5. CRISPR-Cas9 gRNA Design

The three gRNA target regions, exon 2, −3, and −4, were chosen as they are early consecutive exons, with appropriate distance to the ATG start codon in the *HIF1A* gene. The online CRISPR web tools “CRISPor” (http://crispor.tefor.net accessed on 10 October 2020) [32] and “CRISPRon” (https://rth.dk/resources/crispr/crispron/ accessed on 10 October 2020) [33,34] were used to design and evaluate the gRNAs for the CRISPR-Cas9 system, SpCas9 (*Streptococcus pyogenese* CRISPR associated protein 9). The sequence of the three target regions’ exons was set as input in the web tools and submitted with default settings. The final three gRNAs targeting three different exons of *HIF1A* were all chosen by analyzing the CRISPRon/CRISPor output tables, by choosing the gRNA with the highest predicted efficiency (Figure 2a, Appendix A). The gRNA was purchased at Synthego, US. In addition, the robustness of HIF1A gRNA 1 editing was validated by purchasing the gRNA from two different vendors: Synthego.com and IDTdna.com (Integrated DNA Technologies (IDT)).

### 2.6. Evaluation of CRISPR-Cas9 Gene Editing Efficiencies on HIF1A in HUVECs

The CRISPR-Cas9 editing efficiencies for the *HIF1A gene* were analyzed by nucleofecting HUVECs with three different HIF1A gRNAs, as described in the nucleofection protocol above with few alterations. Before nucleofection of the HUVECs, the ribonucleoprotein (RNP) complex was prepared. The synthesized gRNAs (Synthego, Redwood City, CA, USA) were dissolved to be 3.2 µg/µL in nuclease-free water mixed by vortexing and stored at −20 °C. The RNP complex was prepared in three groups: ① HIF1A gRNA 1 + spCas9 protein, ② HIF1A gRNA 2 + spCas9 protein, and ③ HIF1A gRNA 3 + spCas9 protein. All three groups were prepared by mixing 1.8 µL gRNA (3.2 µg/µL), 1.8 µL spCas9 Nuclease V3 (IDT, #1081059), and 3.6 µL nuclease-free water in PCR-tubes and kept at room temperature for 10–60 min. The sample groups in this experiment were: wild type (untreated), 3.2 µg EGFP mRNA treatment (positive control), HIF1A gRNA 1 treatment, HIF1A gRNA 2 treatment, and HIF1A gRNA 3 treatment. The HUVECs sample groups were resuspended in 60 µL OptiMem (Thermo Scientific, #31985062) and 2.4 µL of each RNP complex was added to the appropriate groups. This was followed by nucleofection and 48 h of incubation, in penicillin/streptomycin free medium, before genotyping. The EGFP fluorescence was visualized by a Leica Dmi1 fluorescent microscope to ensure efficient nucleofection after 24 h of incubation. This experiment was performed with one HUVECs donor at p. 4 (*n* = 3). Experiments with HIF1A gRNA 1 were repeated to achieve three biological replicates at p. 2–4, each with technical triplicates.

Genotyping: The HUVECs were washed twice with PBS and the cell pellet was lysed with 200 µL of the lysis master mix (905 µL lysis buffer CS (KCL 50 mM, MgCl_2_ 1.5 mM, Tris/HCL pH8.5 10 mM) + 25 µL Tween20, 20% (Sigma, #P9416) + 50 µL NP-40 Surfact-Amps Detergent (Thermo Scientific, #VK309256) + 20 µL proteinase K (Roche, Basel, Switzerland, #50844900), and collected in PCR strips. The lysed HUVECs were incubated at 65 °C for 30 min and 95 °C for 15 min. The cell lysate was stored at −20 °C. PCR was used to amplify the DNA region of interest before Sanger sequencing.

PCR with PFX polymerase and betaine: All reagents were thawed on ice and vortexed firmly before use. The 20 µL PCR reaction were setup by mixing 2 µL 10× AccuPrime PFX Reaction Mix (Invitrogen, Waltham, MA, USA, #92008) + 1.2 µL forward primer, 5 µM, (Merck, DE) + 1.2 µL reverse primer, 5 µM, (Merck, DE) + 0.16 µL AccuPrime PFX DNA polymerase (Invitrogen, # 92008) + 4 µL betaine (5M) (Sigma, #B0300) + 9.44 µL ddH_2_O + 2 µL cell lysate. See Appendix A for primer details. The Veriti 96well Thermal cycler (Applied Biosystems, Waltham, MA, USA) was used to run the PCR with the following conditions: initial denaturation step 95 °C for 2 min, followed by 45 cycles of denaturation (95 °C for 15 s), annealing (56 °C for 30 s), and extension (68 °C for 30 s). The cycles were followed by a final extension step of 68 °C for 5 min and maintained at 4 °C ∞. The PCR product was visualized by 0.1% agarose gel electrophoresis (Figure 2b). The NucleoSpin Gel and PCR Clean-up kit (Macherey, #2006/001) was used to purify the PCR product. The DNA concentrations and purity of each PCR product were measured by a Nanodrop 1000 Spectrophotometer (Termo Scientific, Waltham, MA, US). The DNA PCR samples were stored at −20 °C.

Sanger sequencing: The PCR products were sent to Sanger sequencing at Eurofins Genomics, DK, by following the manufacturer’s protocol to Mix2Seq Kit (Eurofins Genomics, DK). The Sanger sequencing results were analyzed by Snap gene viewer and the web tool “ICE” (https://ice.synthego.com/#/ accessed on 8 December 2020) was used to calculate the overall CRISPR gene editing efficiency and determine the profiles of all different types of edits present in the Sanger sequencing data. Prism 9, US, was used to plot the gene editing efficiency for each HIF1A gRNAs ± the standard deviation (SD).

### 2.7. FC Analysis of HIF1A KO and WT HUVECs

HIF1A KO and WT HUVECs were prepared for FC analysis as described in the nucleofection protocol, with few alterations: 1/3 were left unstained, and 2/3 of the cells were stained with mouse anti-CD31 FITC (BD Bioscience, #555445) and mouse anti-CD45 BV421 (BD Bioscience, #563879) for 30 min on ice in the dark, After removing debris and doublets, density plots were used to gate the CD45-CD31+ stained HUVECs, (Appendix A).

### 2.8. LDL Uptake Assay

8-well chamber slides (Lab-Tek, #177445) were coated with 0.1% gelatin. HIF1A KO and WT HUVECs were seeded (1·10^5^ cells per replicate (*n* = 3) in 50% M199 medium with supplements and 50% endothelial cell growth medium and incubated for 24 h in a 5% CO_2_ atmosphere at 37 °C. The culturing medium was replaced to M199 medium with supplements and treated with 10 µg/mL Alexa Fluor 594 conjugated acetylated low-density lipoprotein (LDL) (Life Technologies, #L-35353) and incubated for 4 h in a 5% CO_2_ atmosphere at 37 °C. The cells were washed trice with PBS and fixed with 4% paraformaldehyde (PFA) (CellPath, #03809391) for 15 min and washed trice with PBS for 5 min. At the final wash, the cells were stained with 1:1000 Hoechst for 10 min. Cover slides (Thermo Scientific, #174942) were mounted with 3 µL mounting buffer (Thermo Scientific, #P10144). Microscopy images (5 images/well) were obtained with an Olympus BX63 fluorescent microscope equipped with CoolLED pE-300ultra fluorescence microscopy Illumination System and Sensitive Andor Zyla 5.5 camera using 40× (Plan Fluorite) objective. The same intensity of illumination and the same exposure time settings were used for comparative image acquisition as follows; UV excitation (maximum excitation irradiance at 345 nm wavelength) 20% with 10 ms exposure time (emission maximum at 455 nm), GR excitation (maximum excitation irradiance at: 595 nm wavelength) with 200 ms exposure time (emission maximum at 615 nm). Image visualization and analysis was done in Fiji (ImageJ) v. 2.9.0. The background signal was adjusted based on averaging background values from four areas negative for signal in each channel and subtracting the mean values from the final image (Appendix A).

### 2.9. Staining of HIF1A/Hoechst/Actin

Two 0.1% gelatin coated 8-well chamber slides with HIF1A KO and WT HUVECs (80% confluent) were cultured for 1 h in a normoxic- (21% oxygen) or hypoxic (1% oxygen) environment. The cells were fixed with 4% PFA for 15 min and washed trice with PBS for 5 min. The samples for staining were covered with blocking solution (5% FCS serum, 0.3% TritonX-100 in PBS) and 1:100 Human/Mouse/Rat HIF1A antibody (RND Systems, #AF1935), and incubated at 4 °C overnight on plate shaker. The secondary antibody control samples were left untreated in the blocking solution. All the samples were washed twice with PBS for 5 min and resuspended in blocking solution with the secondary antibody 1:500 Alexa flour 594 Donkey anti goat IGG (Life technologies, #A11058). The samples were incubated in the dark on a plate shaker for 2 h at room temperature. All the samples were washed trice with PBS for 5 min. At the final wash the cells were stained with 1:40 Alexa Fluor 488 Phalloidin (actin) (Invitrogen, # A12379) in PBS and incubated for 20 min at room temperature. Followed by 1:1000 Hoechst staining for 10 min in the dark. Cover slides were mounted with 3 µL mounting buffer. Image acquisition was performed (5 images/well) with a Zeiss LSM800 laser scanning confocal microscope, 63X oil objective equipped with diode lasers and three GaAsP detectors. The same laser intensity photo-detector sensitivity and exposure time were applied for comparative image acquisition as follows; lasers: 405 nm: 4%, 488 nm: 5%, 561 nm: 16% with the exposure time of 930.91 ms. The background signal was adjusted based on averaging background values from four areas negative for the signal in each channel and subtracting the mean values from the final image. To assess HIF1A expression in the HIF1KO and WT HUVECs, maximum intensity z-projections of the Hoechst and actin channels were used. ROIs of the nuclei were acquired, and a mask was subsequently created and subtracted from the maximum projection of the actin channel to eliminate the areas comprising the nuclei. In the resulting image, the wand tool was used to trace all areas negative for actin signal and the derived mask was inverted to create a ROI comprising the cell cytoplasm. The mean fluorescence intensity in a grey scale sum projection of the HIF1A channel was calculated using the mean grey values from the nuclei and actin ROIs, respectively. Image analysis was done in Fiji (ImageJ) v. 2.9.0.

### 2.10. Tube Formation Assay

Endothelial cell tube formation assay (TFA) was performed by thawing the Geltrex LDEV-Free Reduced Growth Factor Basement Membrane Matrix (Gibco, #A1413201) in an ice bath at 4 °C overnight. Four 8-well chamber slides were pre-cooled on ice, to avoid the Geltrex Matrix Solution (GMS) from immediately solidifying. The GMS was mixed by gentle pipetting and kept on ice until added, 100 µL/cm^2^, and evenly distributed in the wells. The GMS was allowed to solidify for 30 min at 37 °C. Every 5 min the slides were gently tapped against the flow bench table to minimize the formation of a strong concave meniscus. When the GMS had solidified, 0.3·105 cells (HIF1A KO and WT HUVECs) were seeded per well in 50% M199 medium with supplements and 50% endothelial cell growth medium and incubated for 24 h in a 5% CO_2_ atmosphere at 37 °C. Finally, the TFA HIF1A KO and WT HUVECs were incubated in a normoxic or hypoxic environment for 2 h. The cells were fixed with 4%PFA for 15 min and washed trice in PBS. Bright field images (6 images/well) were taken with a Leica DMi1 microscope, DE. Image visualization and analysis was done in Fiji (ImageJ) v. 2.9.0. The background signal was adjusted based on averaging background values from four areas negative for signal in each channel and subtracting the mean values from the final image. The tube formation analysis program WimTube from Wimasis.com, ES, was used to make a quantitative analysis of the six TFA images. Gaphpad Prism 9, US, was used to plot the WimTube analysis data (mean ± SD) of the HIF1A KO and WT HUVECs cultured in both normoxic and hypoxic conditions. One-way ANOVA with Tukey multiple comparisons test compared each sample group to the WT HUVECs cultured in normoxic conditions.

## 3. Results

### 3.1. Efficient mRNA Delivery into HUVECs by Nucleofection

We first sought to establish an efficient nucleofection protocol for HUVECs. To accurately quantify transfection efficiency and efficacy, we used an in vitro transcribed (IVT) mRNA encoding an Enhanced Green Fluorescent Protein (EGFP). The expression of EGFP in the HUVECs was quantified by fluorescence microscopy and flow cytometry (FC) analysis. High concentrations of the transfection reagent could cause cellular toxicity [35]. Hence, we tested different amounts of EGFP mRNA to quantify the optimal concentration of EGFP mRNA for nucleofection of HUVECs. The nucleofection protocol was established by nucleofecting HUVECs with different amounts of EGFP mRNA (0.4, 0.8, 1.6, 3.2, and 4.8 µg, *n* = 3 per group). The bright field images showed that the morphology and confluency of the HUVECs after nucleofection with mRNA, appeared normal with few dead cells in all groups. The HUVECs receiving 4.8 µg mRNA were less confluent compared to those receiving lower doses (Figure 1a), which indicated a dose-dependent negative impact on ECs growth. Fluorescence microscopy showed that green fluorescence was visible from 0.8 µg and increased accordingly with the increase of EGFP mRNA (Figure 1a). FC analysis showed that the majority (74%) of the nucleofected HUVECs were EGFP positive when nucleofected with 0.4 µg IVT EGFP mRNA (Figure 1b,c), suggesting that the FC analysis was more sensitive than the fluorescence microscopy analysis. The fraction of EGFP-positive cells increased significantly from 74% to 96% when increasing EGFP mRNA to 0.8 µg per nucleofection, and nearly 100% of the cells were EGFP positive when nucleofected with EGFP mRNA concentrations from 0.8–4.8 µg (Figure 1c). We next quantified the expression level of the EGFP in the nucleofected HUVECs. The median fluorescence intensity (MFI) of EGFP-positive cells increased as more EGFP mRNA was used per nucleofection (Figure 1d). Although nearly 100% transfection efficiency was already achieved with 0.8 µg mRNA, the efficacy of gene expression still significantly increased with the amount of mRNA used per nucleofection (Figure 1c,d). We also validated the robustness of transfection efficiency and EGFP expression by mRNA as compared to traditional DNA plasmid-based delivery (Appendix A). We observed less confluent HUVECs upon nucleofection with 4.8 µg IVT EGFP mRNA, indicating a potentially negative effect on the primary ECs growth (Figure 1a), thus 3.2 µg EGFP mRNA was used for further studies.

### 3.2. Efficient CRISPR Gene Editing of Primary HUVECs

We next tested whether the established nucleofection and mRNA delivery approach could be used to achieve efficient CRISPR gene editing in primary HUVECs. To this end, we used a pre-formed ribonucleoprotein (RNP) complex comprising the SpCas9 protein and chemically modified synthetic guide RNA (gRNA). Three gRNAs were designed to target the early consecutive exons of the *HIF1A* gene (Figure 2a). Sanger sequencing (Figure 2b) and ICE-based indel deconvolution analysis showed that high gene editing efficiencies were achieved in primary HUVECs with all three gRNAs: 98% for gRNA 1, 79% for gRNA 2, and 66% for gRNA 3 (Figure 2c). The highly efficient CRISPR gene editing efficiency of gRNA 1 is really striking. Unlike the other two gRNAs (2 and 3), HIF1A gRNA 1 only creates an indel of thymine (T) insertion (Figure 2d), which leads to the introduction of a stop codon (Appendix A). The dominant indel type of T insertion at the double-brand break site corroborates our previous observation of the CRISPR 1bp insertion indel profiles [33]. Furthermore, we tested synthetic gRNA (HIF1A gRNA 1) provided by two different vendors (Synthego and Integrated DNA Technologies (IDT)), which both resulted in an efficiency of nearly 100% and the same indel formation (Appendix A). This unique indel formation is consistent with our previous indel profiling using self-targeting surrogate libraries [33]. Notably, the CRISPR efficiency achieved in our study was based on a completely selection-free setting. This confirms that the nucleofection and RNP delivery approach can be used to achieve highly efficient CRISPR gene editing in primary ECs.

**Figure 2 biomolecules-13-00023-f002:**
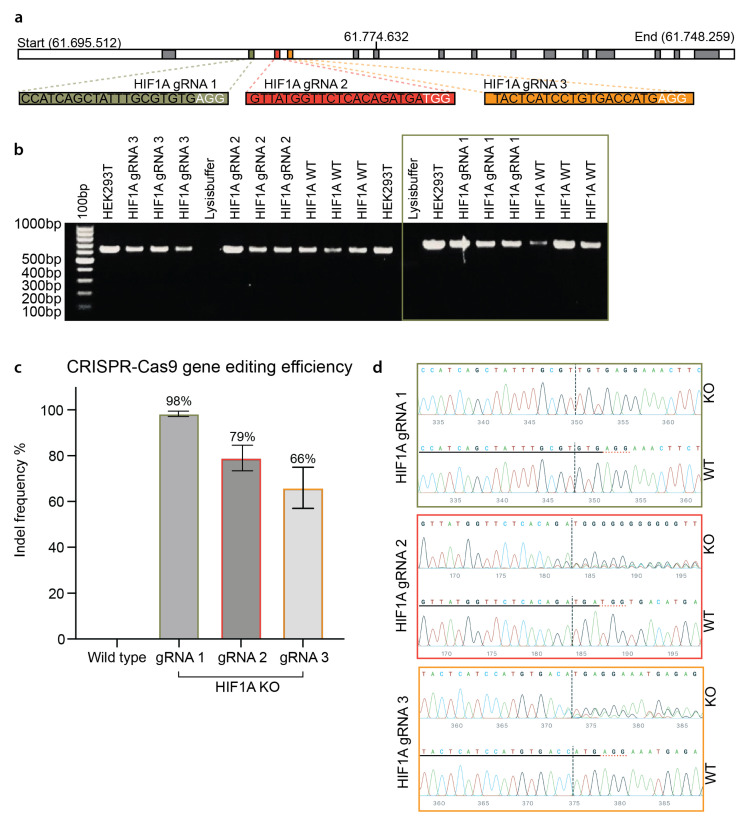
CRISPR gene editing of the *HIF1A* gene in HUVECs. (**a**) Graphical illustration of the *HIF1A* gene, target exons, and gRNAs. The top panel shows the length of the *HIF1A* gene (chromosome 14 at position 61.695.512–61.748.259). Three isoforms of the gene are known, which contain 14–15 exons (indicated by the grey boxes). The green, red, and yellow, boxes visualize the three target consecutive exons (exon two, −three, and −four). The sequence of HIF1A gRNA 1, −2, and −3 is captured in green, red, and yellow bars. (**b**) PCR amplification of the CRISPR gene editing of HUVECs. Visualization of the gel electrophoresis of the amplified DNA from HIF1A gRNA 1, −2, and −3 gene edited HUVECs, HIF1A WT (wild type) HUVECs, HEK293T cells, and the lysis buffer. The 100 bp gene marker is shown in the first well. The green frame indicates the samples amplified by the HIF1A F1/R1 primer sets. (**c**) CRISPR gene editing efficiency of the three different HIF1A gRNAs. Quantification of the gene editing efficiency of the HIF1A gRNA 1, −2, and −3 and the WT. Values are presented as the mean percentage of indels ± STD. (**d**) Representative visualization of the Sanger sequencing results for each HIF1A gRNA. For each HIF1A gRNA, the WT HUVECs sequence is shown in the bottom row. The gRNA sequence is underlined with black. The PAM sequence is underlined in dotted red. The editing site is marked by the black dotted vertical line. The sequence of the gene-edited part is shown by wave plots. The nucleotides are color coded as G = guanine, C = cytosine, T = thymine, and A = adenine. (**b**–**d**) Experiments on HUVECs from three biological donors at p. 2–4 (*n* = 3). HIF1A gRNA 1–3 experiments are performed with one biological donor at p. 4. HIF1A gRNA 1 experiments were repeated to achieve three biological replicates.

### 3.3. Functional Validation of HIF1A KO HUVECs

To validate if HIF1A gRNA 1 had successfully disrupted gene expression at the protein level, we cultured the HIF1A gRNA 1 KO (hereafter referred to as HIF1A KO) and wild type (hereafter referred to as WT) HUVECs in normoxic (21% oxygen) or hypoxic (1% oxygen) conditions and assessed HIF1A expression by antibody-based protein staining (Figure 3). We had tested different hypoxic cultivation times (1, 2, and 4 h), of which 1 h hypoxic cultivation could clearly induced HIF1A nucleus translocation in ECs without triggering massive cell death. In WT HUVECs, we observed a clear upregulation and translocation of the HIF1A protein from the cytoplasm (normoxic, Figure 3a,e) to the nucleus (hypoxic, Figure 3b,e), commensurate with the HIF1A function and pathway (Appendix A) [36,37,38,39,40,41]. In the HIF1A KO cells, no expression of HIF1A was detected in neither normoxic nor hypoxic conditions (Figure 3c–e). We also confirmed the expression of CD31 and functional uptake of LDL of the HIF1A KO and WT HUVECs, as confirmation of their EC phenotype (Appendix A).

We next sought to evaluate if HIF1A KO affects ECs functions. Tube formation assay (TFA) is a commonly used method to study angiogenesis [42]. We performed TFA to investigate if HIF1A KO affects HUVECs’ ability to form capillary-like structures (*tubes*). Both HIF1A KO and WT HUVECs cultured in normoxic or hypoxic conditions can form a tube-like network. WT HUVECs formed tube-like networks with long branches, more tight junctions, and a clear mesh pattern under both conditions (Appendix A). In contrast, the HIF1A KO HUVECs in normoxic conditions showed a weaker ability to form the tube-like network and fenestrated junctions (Appendix A). This difference was more pronounced when cultured in hypoxic conditions. The tube-like network of HIF1A KO HUVECs cultured in hypoxic conditions was disrupted (Appendix A). WimTube analysis was further performed to make a quantitative analysis of the TFA images (Figure 4a). The tube formation process was investigated by measuring the total tube length in pixels (px), the total number of tubes (count), mean tube length (px), percentage of covered area, and total branching points (counts) (Figure 4). Quantitative results showed that there was no significant difference between the HIF1A KO and WT HUVECs in total tube lengths (Figure 4b). The HIF1A KO HUVECs in both normoxic and hypoxic conditions had a significantly increased number of tubes (Figure 4c), and significantly decreased mean tube length (Figure 4d). In addition, the number of branching points was significantly increased in HIF1A KO HUVECs in both normoxic and hypoxic conditions (Figure 4e). The HIF1A KO HUVECs cultured in hypoxic conditions showed challenged angiogenesis (Appendix A), which was quantitively confirmed as the percentage of tube-covered area was significantly decreased compared to the WT HUVECs (Figure 4f). In conclusion, we confirmed that *HIF1A* deficiency affects HUVECs ability of angiogenesis and response to hypoxia.

## 4. Discussion

Primary ECs models, like HUVECs, most closely represent the tissue of origin compared to secondary or immortalized cell lines, but are challenging as they are hampered by a limited life span, low transfection efficiencies, and high contamination risks [18,19,20]. Thus, previous gene editing studies have mainly been performed on immortalized ECs [21,22]. This study created a protocol for highly efficient RNP-mediated CRISPR gene editing of primary ECs (HUVECs), targeting the *HIF1A* gene, which to our knowledge has not been published to date.

First, an efficient protocol for nucleofection of HUVECs with EGFP mRNA was established, which resulted in transfection efficiencies of nearly 100%. Previous studies like Moradian et al. transfected primary cells with EGFP mRNA, which resulted in 70% EGFP positive macrophages, quantified by FC, with no significant change in the cell viability [43]. Hunt et al. investigated different transfection reagents to transfect HUVECs with EGFP plasmid, the most efficient transfection reagent resulted in 49% of cells expressing EGFP [44]. The microscopy images (Appendix A) show that the HUVECs transfected with EGFP mRNA were more confluent compared to the EGFP plasmid transfected HUVECs. This indicates that EGFP mRNA nucleofection is less toxic to the HUVECs compared to EGFP plasmid. One of the main advantages of mRNA nucleofection, in contrast to plasmid nucleofection, is that it avoids transcription and is less toxic as it results in transient gene expression -since the mRNA is less stable [43].

The gene editing delivery method has a great impact on the gene editing efficiency of the primary ECs, thus the optimization of the nucleofection protocol has been of great importance for our experiments. Previous studies have used lentiviral vector mediated CRISPR as a delivery system and report gene editing efficiencies of 40–58% [10,25]. Gong et al. report gene editing efficiencies of 40–60% by dual viral vector (lenti/adeno virus) mediated CRISPR gene editing [24]. The most efficient CRISPR gene editing of primary ECs, to our knowledge [15], resulted in 80% gene disruption by AAV5 mediated CRISPR. In 2019 Schwefel et al. compared lentiviral and RNP-mediated CRISPR gene editing on immortalized HUVECs, which resulted in 66% lentiviral and 63% RNP gene editing efficiencies [22]. The RNP-mediated CRISPR gene editing is also highly efficient in other primary cell types like human primary T cells [28] and B cells [45]. The RNP-based method has many advantages, as it enables immediate gene editing, and has a short presence of Cas9 protein, since the protein is degraded. This results in specific gene editing with few off-targets and low toxic effects [46]. These advantages founded our opinion that RNP-mediated CRISPR will be efficient for gene editing of primary ECs.

This study accomplished the creation of a highly efficient protocol for RNP-mediated CRISPR gene editing of primary ECs, with a gene editing efficiency of 98% of the *HIF1A* gene in primary HUVECs. The *HIF1A* gene is well studied, found to control transcription of over 40 genes, playing an important role in endothelial adaption, vascular development, and angiogenesis [37,38]. The HIF1A gRNA 1 showed a remarkably high gene editing efficiency of 98% and repaired only by the insertion of one thymine at the CRISPR-induced double strand break site. The consistent one base pair insertion occurs 17 bp upstream of the PAM sequence, which confirms our previous findings that reveal how one base pair insertions most frequently result in the insertion of the same nucleotide as N17 upstream of the PAM [33]. This might be related to the NHEJ mechanism, as it prefers one base pair insertion after the CRISPR induced double stranded breaks since this will be one of the fastest repair options [33,47].

Our results show that the HIF1A KO results in impaired angiogenesis, as we see many branching points, short tube lengths, and less tube covered area. The impaired angiogenesis might be caused by an imbalance in the proportion of tip and stalk cells, thus resulting in “split ends” in the sprouting of the ECs. Similarly, Tang et al. deleted *HIF1A* in ECs by cross-breeding Tie2-Cre transgenic mice and found that deletion of *HIF1A* in primary murine lung ECs disrupt vascular endothelial growth factor (VEGF)-dependent signaling pathway in vivo, which resulted in impaired angiogenesis [48]. Impaired angiogenesis in diabetes complications, like diabetic retinopathy, is treated by anti-angiogenic therapy (AAT), which targets the tip cells in the sprouting ECs by antagonizing the VEGF receptor [49,50,51]. Unfortunately, AAT requires regular injections, and some patients acquire resistance to the AAT as the ECs adapt their angiogenic mechanisms [52,53]. New model organisms elucidating VEGF, angiogenesis, and ECs functions are needed to develop better VEGF targeting AAT. Promisingly, Holmgaard et al. demonstrate that RNP-mediated CRISPR effectively generates a VEGF KO in mice retina, which is a potential strategy for future treatment of retinal diseases, but further studies are needed [54]. Although not investigated in our study, the efficient approach of biallelic and selection-free CRISPR gene (*HIF1A*) knockout in primary ECs allows us to investigate how gene (*HIF1A*) disruption affects the EC transcriptional machineries, functions, plasticity, and heterogeneity using, e.g., single cell RNA sequencing.

In summary, we established an RNP-mediated CRISPR gene editing protocol for primary ECs allowing extremely efficient HIF1A KO in primary HUVECs. The RNP-mediated CRISPR gene editing of *HIF1A* in primary ECs, resulted in gene editing efficiencies up to 98%. HIF1A gRNA 1-based editing created a one base pair insertion leading to an early stop codon. The functional validation assays show that the HIF1A KO HUVECs are functional ECs as they have an uptake of LDL and express CD31 but the KO results in insufficient angiogenesis.

## Figures and Tables

**Figure 1 biomolecules-13-00023-f001:**
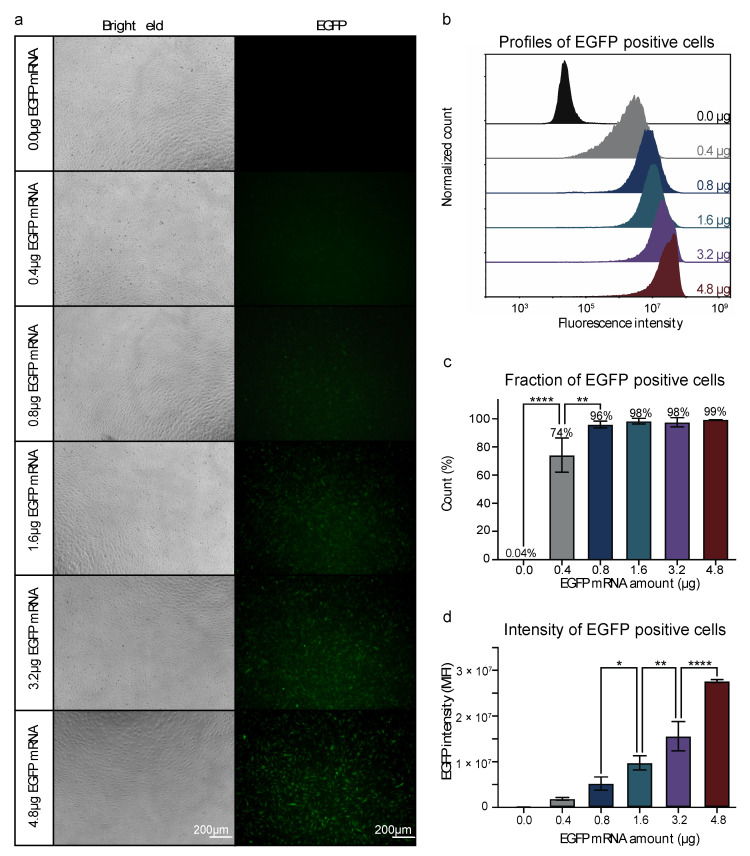
Nucleofection of HUVECs with a titration of EGFP mRNA. (**a**) Visualization of the EGFP positive cells by fluorescent- and bright field microscopy. (**b**) The profiles of EGFP-positive HUVECs at each EGFP mRNA concentration. Values are presented as normalized counts of the fluorescent intensity. (**c**) Fraction of EGFP-positive HUVECs at each EGFP mRNA concentration. Values are presented as the mean percentage of EGFP-positive cells ± SD. Neighboring EGFP mRNA concentrations are compared by one-way ANOVA with Tukey multiple comparisons test, ** *p* ≤ 0.01, **** *p* ≤ 0.0001. (**d**) Median fluorescence intensity of EGFP-positive cells at each EGFP mRNA concentration. Values are presented as the mean ± SD. Neighboring EGFP mRNA concentrations are compared by one-way ANOVA with Tukey multiple comparisons test, * *p* ≤ 0.05, ** *p* ≤ 0.01, **** *p* ≤ 0.0001. (**a**–**d**): Experiments on HUVECs from one biological donor at p. 5 (*n* = 3).

**Figure 3 biomolecules-13-00023-f003:**
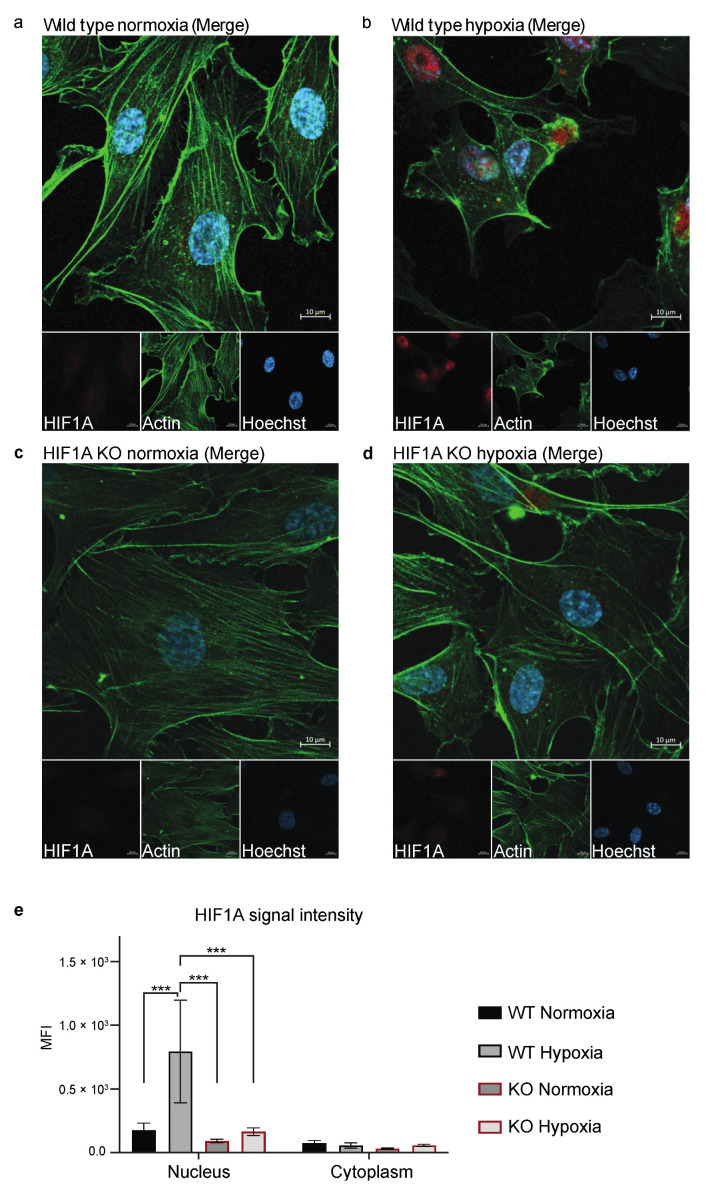
Staining of HIF1A in HIF1A KO and WT HUVECs. (**a**–**d**) One representative image of the stained HIF1A KO and WT HUVECs, which were stained with Hoechst (blue), anti-actin (green), and anti-HIF1A (red). (**a**) WT HUVECs cultured in normoxic conditions. (**b**) WT HUVECs cultured in hypoxic conditions. (**c**) HIF1A KO HUVECs cultured in normoxic conditions. (**d**) HIF1A KO HUVECs cultured in hypoxic conditions. (**e**) Quantification of the HIF1A signal intensity (MFI) in the nucleus and the cytoplasm in WT and KO HUVECs in normoxic and hypoxic conditions, which is compared by two-way ANOVA multiple comparisons test, *** *p* ≤ 0.001. The experiment was performed with one biological replicate at p. 3 (*n* = 3) and five images were taken per well. Scale bar 10 µm.

**Figure 4 biomolecules-13-00023-f004:**
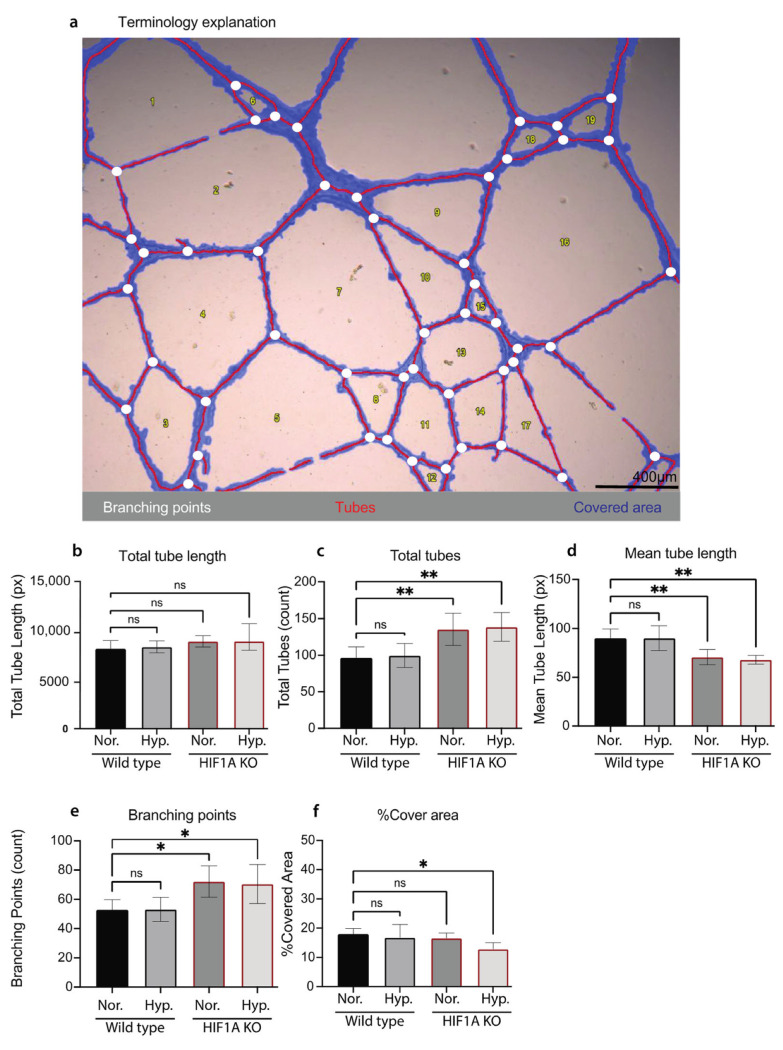
Quantification of the tube formation assay of HIF1A KO and WT HUVECs. (**a**) One representative bright field image of the TFA HUVECs, visualizing the Wimasis terminology which illustrates the tubes in red, the branching points in white, and the covered area in blue. Scale bar 400 µm. (**b**–**f**) Quantification of the TFA analysis of HIF1A KO and WT HUVECs cultured in normoxic conditions (Nor.) or 2 h hypoxic conditions (Hyp.). (**b**) The total tube length calculated in pixels. (**c**) The total amount of tubes, (count). (**d**) The mean tube length, (pixels). (**e**) Shows the total branching points, (count). (**f**) The percentage of covered area. The mean of each test ± STD is plotted, and one-way ANOVA with Tukey multiple comparisons test compared each sample group to the WT HUVECs cultured in normoxic conditions (ns = no significance, * *p* ≤ 0.05, and ** *p* ≤ 0.01). (**a**–**f**): Experiments on HUVECs from one biological donor at final p. 3.

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
