# Peer review of "HIF1A Knockout by Biallelic and Selection-Free CRISPR Gene Editing in Human Primary Endothelial Cells with Ribonucleoprotein Complexes"

_biomolecules, 2022, doi:10.3390/biom13010023_

Round 1
Reviewer 1 Report
The manuscript is well-written, contains a thorough description of Methods, and addresses a significant issue of efficient RNP-mediated CRISPR gene editing protocol for primary endothelial cells. The protocol may help improving the experimental arsenal available for research in endothelial cells. I have no major concerns but a few notes for authors to re-address and comment.
1. For comparing fluorescence signals, the same laser intensities and camera exposures have to be applied. There are no clear notes in this regard. If true, this has to be specified in Methods and the Fig1 legend.
2. The authors write that HIF1A gRNA1 creates an indel of thymine (T) deletion (lines 353, 516). However, in Fig2d and FigS2, the provided KO sequences contain one extra T base as compared to the wild type sequence. This additional T base shifts the open reading frame to introduce an early stop codon 24 bp downstream. Why is this called deletion but not insertion like it is done in line 489? Please, comment.
3. Lines 139 and 228, ‘fitch microscopy images’ and 'anti-CD31 FITCH’, respectively. Did you mean FITC?
4. Line 478 reads ‘human primary T cells87’. What does 87 stay for?
Author Response
The manuscript is well-written, contains a thorough description of Methods, and addresses a significant issueof efficient RNP-mediated CRISPR gene editing protocol for primary endothelial cells. The protocol may helpimproving the experimental arsenal available for research in endothelial cells. I have no major concerns but a few notes for authors to re-address and comment.
Response: We really appreciate the reviewer’s valuable comments and the recommendation of our work. In this revision, we have thoroughly addressed the comments.
- For comparing fluorescence signals, the same laser intensities and camera exposures have to be applied.There are no clear notes in this regard. If true, this has to be specified in Methods and the Fig1 legend.
Response: Yes, for each experiment involving fluorescence images, we have used the same setting for image capture. In the revised manuscript line 140, we have specified this information.
Specific setting for the confocal microscopy has also been included in the method. Line 248-260. “Images (5 images/well) were obtained with an Olympus BX63 fluorescent microscope equipped with CoolLED pE-300ultra fluorescence microscopy Illumination System and Sensitive Andor Zyla 5.5 camera using 40x (Plan Flurite) objective. The same intensity of illumination and the same exposure time settings were used for comparative image acquisition as follows; UV excitation (maximum excitation irradiance at 345nm wavelength) 20% with 10ms exposure time (emission maximum at 455nm) , GR excitation (maximum excitation irradiance at: 595nm wavelength) with 200 ms exposure time (emission maximum at 615nm).”
Line 285-287. “A Zeiss LSM800 laser scanning confocal microscope, 63X oil objective equipped with diode lasers and three GaAsP detectors. The same laser intensity photo-detector sensitivity and exposure time were applied for comparative image acquisition as follows; lasers: 405nm: 4%, 488nm: 5%, 561nm:16% with the exposure time of 930.91ms.”
- The authors write that HIF1A gRNA1 creates an indel of thymine (T) deletion (lines 353, 516). However, in Fig2d and FigS2, the provided KO sequences contain one extra T base as compared to the wild type sequence. This additional T base shifts the open reading frame to introduce an early stop codon 24 bp downstream. Why is this called deletion but not insertion like it is done in line 489? Please, comment.
Response: Really appreciate the reviewer’s correction for the term here. It was a mistake in describing the indel. We have corrected the term thoroughly.
- Lines 139 and 228, ‘fitch microscopy images’ and 'anti-CD31 FITCH’, respectively. Did you mean FITC?
Response: Thanks for the correction, it should FITC. FITCH was meant the FITC-H (height). We have corrected the term.
- Line 478 reads ‘human primary T cells87’. What does 87 stay for?
Response: This was an error for the reference. We have corrected in the revision. Great thanks for the careful reviewing.
Reviewer 2 Report
This is very intresting manuscript providing technical details for HIF1A KO. The results are solid, the flow logical and no serious flaws detetcted. I have just minor comments:
1. Authors should consider testing how the knockout affected the transcriptional activity of HIF1 - this may be helpful: https://doi.org/10.1096/fj.202101987R and10.1096/fj.201802650RR
2. The HIF1 IN HUVECs is present untill 8 h of hypoxia exposure (see https://doi.org/10.1186/s11658-022-00408-7) so more elaboration why 2 hours where used would be helpfull
3. It would be good to check how the KO affected HIF2alpha levels.
Author Response
This is very intresting manuscript providing technical details for HIF1A KO. The results are solid, the flow logical and no serious flaws detetcted. I have just minor comments:
Response: We really appreciate the reviewer’s work in reviewing our manuscript and the constructive and valuable comments.
- Authors should consider testing how the knockout affected the transcriptional activity of HIF1 - this may behelpful: https://doi.org/10.1096/fj.202101987R And 10.1096/fj.201802650RR
Response: Thanks for the suggestion. This is indeed great suggestions and experiments that we should consider investigating how HIF1 knockout affect the transcriptome in the endothelial cells. Since the manuscript is more focusing on sharing this highly efficient CRISPR gene editing protocol in endothelial cells with the scientific community, validations comprising Sanger sequencing, immunofluorescent staining, and TFA assays are sufficient to support the CRISPR disruption of HIF1A. We believe the reviewer agree with us that a deeper investigation of how HIF1 knockout affect HIF1 regulon activity and endothelial cell response using e.g., single cell RNA sequencing will be valuable follow-up studies. We had included some elaborations in the revised discussion (line 564-568).
- The HIF1 IN HUVECs is present untill 8 h of hypoxia exposure (see https://doi.org/10.1186/s11658-022-00408-7) so more elaboration why 2 hours where used would be Helpful.
Response: Thanks for the comment to the hypoxia exposure and the reference. When setting up the hypoxia condition, we did have tested 1, 2, 4 hours exposure. The ECs experience great stress when hypoxia exposure is longer than 1 hour, and thus hampering the TFA formation. We elaborated the selection of the hypoxia condition in the revised manuscript line 434-436.
- It would be good to check how the KO affected HIF2alpha levels.
Response: Great thanks for the suggestion. From an endothelial cell biology point of view, it is a great idea to evaluate how does HIF1A deficiency in primary ECs affect the expression the HIF family proteins, as well as the downstream transcription changes. As the HIF1A transcriptional regulation and network is a little bit out of the key focus of the current manuscript, the thorough investigation of how HIF1A affect endothelial cell transcription and phenotype, particularly using single cell RNA sequencing, will be carried out in our follow-up studies.